# Population Older than 69 Had Similar Fatality Rates Independently If They Were Admitted in Nursing Homes or Lived in the Community: A Retrospective Observational Study during COVID-19 First Wave

**DOI:** 10.3390/geriatrics8030048

**Published:** 2023-04-28

**Authors:** Javier Martínez-Redondo, Carles Comas, Cristina García-Serrano, Montserrat Crespo-Pons, Pilar Biendicho Palau, Teresa Vila Parrot, Francisco Reventoz Martínez, Lídia Aran Solé, Neus Arola Serra, Eva Tarragona Tassies, Jesús Pujol Salud

**Affiliations:** 1Balaguer Primary Care Center, Institut Català de la Salut (ICS), 25600 Lleida, Spain; 2Department of Mathematics, Campus ETSEAFIV, University of Lleida, 25001 Lleida, Spain; 3Research Group in Therapies in Primary Care (RETICAP Group), 25007 Lleida, Spain; 4Biomedical Research Institute (IRB Lleida), University of Lleida (UdL), 25198 Lleida, Spain

**Keywords:** COVID-19 virus infection, mortality, nursing homes, community health care, comorbidity, older people, first wave

## Abstract

The aim of this study is to assess the influence of living in nursing homes on COVID-19-related mortality, and to calculate the real specific mortality rate caused by COVID-19 among people older than 20 years of age in the Balaguer Primary Care Centre Health Area during the first wave of the pandemic. We conducted an observational study based on a database generated between March and May 2020, analysing COVID-19-related mortality as a dependent variable, and including different independent variables, such as living in a nursing home or in the community (outside nursing homes), age, sex, symptoms, pre-existing conditions, and hospital admission. To evaluate the associations between the independent variables and mortality, we calculated the absolute and relative frequencies, and performed a chi-square test. To avoid the impact of the age variable on mortality and to assess the influence of the “living in a nursing home” variable, we established comparisons between infected population groups over 69 years of age (in nursing homes and outside nursing homes). Living in a nursing home was associated with a higher incidence of COVID-19 infection, but not with higher mortality in patients over 69 years of age (*p* = 0.614). The real specific mortality rate caused by COVID-19 was 2.27^0^/_00_. In the study of the entire sample, all the comorbidities studied were associated with higher mortality; however, the comorbidities were not associated with higher mortality in the infected nursing home patients group, nor in the infected community patients over 69 years of age group (except for neoplasm history in this last group). Finally, hospital admission was not associated with lower mortality in nursing home patients, nor in community patients over 69 years of age.

## 1. Introduction

Since the appearance of the first case of COVID-19 in China, in December 2019 [1,2,3,4], the SARS-CoV-2 infection has rapidly spread throughout the world, being declared a pandemic by the WHO on 11 March 2020 [5]. Since then, it has been the main health problem in the world. As of 11 December 2022, over 645 million confirmed COVID-19 cases and over 6.6 million COVID-19-related deaths were reported globally [6].

In Spain, the first confirmed case was reported on 31 January 2020 [7]; in Catalonia, on 25 February 2020 [8]; in Lleida, on 6 March 2020; and in the Primary Care Centre (PCC) of Balaguer, on 13 March 2020. This was the beginning of what would be called the first wave of SARS-CoV-2 in Spain [9,10]. On 13 March 2020, there were 4209 COVID-19 patients [11]; therefore, the government approved the Royal Decree 463/2020 (14 March 2020) and declared a state of alarm for the management of the health crisis [12]. People’s freedom of movement was limited; most shops, and all leisure, educational, and cultural places were closed. These lock down measures caused a decrease in infections and mortality from April to May 2020 [10]. Finally, on 21 June 2020, after more than 250.000 infected patients and 20.000 deaths, the government of Spain declared the conclusion of the state of alarm, putting a definitive end to the first wave of COVID-19 [13].

During the first wave, the fatality rate was 2.16% from the age of 60, 5.24% from the age of 70, and 17.91% from the age of 80 [14]. According to the data from the Ministry of Health, 86% of the total deaths were among people over 70 years of age. The same source reported that, in June 2020, 70.89% of the deceased patients lived in nursing homes [15].

During the first wave of infection in Spain, RT-PCR were not available in extra hospital facilities, and this led to great difficulty in obtaining real epidemiological data on COVID-19-related mortality. Knowing the specific mortality rate for COVID-19 continues to be a challenge for society, and that is why many studies have estimated the real COVID-19-related mortality during the first wave using excess mortality analysis [16,17,18,19].

Many studies were published by epidemiology services, prison health services, and hospital services, and they show the impact of SARS-CoV-2 infection on the population in terms of morbidity and mortality, symptoms, and risk of hospitalization, for example [20,21,22,23,24,25,26,27,28,29,30].

Other studies illustrate the importance of primary care in research related to COVID-19 infection [31,32,33,34,35], as well as the importance of the databases generated from these services in the COVID-19-related mortality studies [19].

In this context, we felt special concern about whether living in a nursing home could be a sufficient variable to explain the high mortality that occurred in those facilities during the first wave of the COVID-19 infection.

In relation with this issue, even though the highest mortality during the first wave of SARS-CoV-2 infection occurred in nursing homes, we only found two studies [19,36] that analyse the influence of the variable “living in a nursing home” on COVID-19-related mortality compared to the population living in the community.

In this context, we aimed to analyse how living in a nursing home (LNH) influenced COVID-19-related mortality and to calculate the real specific mortality rate caused by COVID-19 (including COVID-19-related mortality not confirmed by RT-PCR) among people older than 20 years of age in the health area of Balaguer during the first wave. We considered several risk factors; in particular, we considered whether those people were admitted in a nursing home.

## 2. Materials and Methods

### 2.1. Study Design and Participants

This retrospective observational study was carried out in the Balaguer Primary Care Health Area from 13 March to 28 May 2020. The inclusion criteria comprised being over 20 years of age and being included in the database. We decided to set the cut-off point at 20 years of age to be able to study the entire adult population living in the community, and to be able to compare the results of our secondary aim (calculating the true specific mortality rate) with similar published studies.

On 7 April 2020, the PCC Health Area of Balaguer included 28,047 people, of which 22,458 were older than 20 years of age. In the area, 6 nursing homes were accounted with a total of 406 residents. In mid-March, the nursing homes were totally full.

The final studied sample consisted of 211 patients with a mean age of 70 years, of which 86 (40.75%) were male.

### 2.2. Variables

Data were collected from two different populations, including infected patients living in nursing homes and infected patients living in the community (outside nursing homes). The large difference in age between the two populations, as well as the close relationship between the variables of age, nursing home admission, and COVID-19 mortality [15], made it necessary to add a third study population group (infected community population older than 69 years of age) to establish the real influence of being admitted in a nursing home on COVID-19 mortality and control biases that could be associated with age.

The other study variables were as follows: age; sex; symptoms (temperature above 37 °C, odynophagia, expectoration, cough, headache, dyspnoea, anosmia, asthenia, and diarrhoea); evolution during the month after the confirmation of the infection or the onset of symptoms (cured/deceased); and hospital admission. Comorbidities were collected from the clinical history and included in the database (obesity; hypertension; diabetes mellitus; chronic obstructive pulmonary disease (COPD); cardiovascular disease; chronic kidney failure; neoplasm history; and neurological pathologies).

### 2.3. Statistical Analysis

We calculated absolute and relative frequencies of symptoms and comorbidities, and descriptive measures of centralization, such as the arithmetic mean and the median.

We also generated contingency tables to compare different categorical variables of patients with a confirmed infection of SARS-CoV-2. This way, we could study COVID-19-related mortality according to the different variables of interest.

We calculated the specific mortality rate (patients died from COVID-19/total population per 1000 inhabitants), and the case fatality rate (patients died from COVID-19/infected patients per 100 inhabitants), in total sample and in different groups.

Finally, we performed a chi-square test to assess statistical differences between population groups. *p* < 0.05 was considered statistically significant. Statistical analyses were performed using the R statistical package, version 4.0.2. (R Core Team, 2020).

### 2.4. Data Collection

On 13 March 2020, the PCC Balaguer direction established the respiratory isolation unit (COVID-19 Unit) to serve the population with suspected or confirmed COVID-19 infection. On 16 March 2020, because of the epidemiological emergency and the lack of information on COVID-19, a database was created to collect follow-up data of the patients, their symptoms, comorbidities, and their evolution in association with COVID-19. The main purpose of the continued examination of these data was to improve the effectivity of clinical decisions on the population.

Every 48 h, the workforce assigned at the COVID-19 Unit registered all new cases of COVID-19 in patients belonging to the Balaguer PCC Health Area and included the accumulated daily incidence in the database. They considered patients treated inside the unit, patients attended to at home, and patients visited in nursing homes. In addition, they reviewed all the deaths among patients belonging to the Balaguer PCC Health Area.

Finally, the database collected all the patients with an RT-PCR-confirmed diagnosis of COVID-19 belonging to the Balaguer PCC Health Area, and deceased patients that did not undergo RT-PCR, but whose clinical history showed suspected COVID-19 infection as the cause of death. Specifically, COVID-19 was established as the cause of death when there was a lack of response to antibiotics, all the clinical (fever, cough, dyspnoea, asthenia, etc.) and epidemiological data (e.g., close contact with relatives or other patients in a residence) of the clinical history pointed in that direction, and there was no other triggering cause in the days prior to the onset of symptoms, or no other suspected cause of death.

## 3. Results

### 3.1. Sample Distribution

The final studied sample consisted of 211 patients with a mean age of 70 years, of which 86 (40.75%) were male.

Of the 211 patients, 108 (51.18%) were admitted in nursing homes, the mean age was 86 years, and 42 (38.89%) were male. The other 103 patients (48.82%) were living in the community, with the mean age of 53.78 years, and 44 (42.71%) were male. Of the community patients, 25 (24.27%) were older than 69 years of age, with a mean age of 79.04 years, and 18 (72%) of them were male (see Table 1).

For 204 (96.68%) of the 211 patients, an RT-PCR test was obtained.

A total of 51 (24.17%) patients died, 44 of them (86.27%) with a diagnosis of COVID-19 confirmed by RT-PCR, and 7 (13.72%) with a diagnosis of suspected COVID-19. Of the latter, five (9.80% of total mortality) were admitted in nursing homes and two (3.92% of total mortality) lived in the community.

Of the total deceased patients, 41 (80.39%) were admitted in nursing homes. Of these, 40 (97.56%) were over 69 years of age. The other 10 (19.60%) deceased patients lived in the community. Of these, 8 (15.68% of total mortality) were over 69 years.

### 3.2. Symptoms and Comorbidities Distribution

The most frequent symptoms were fever, dyspnoea, cough, and asthenia in all groups studied. The community patients presented more frequency of cough (*p* = 0.001), headache (*p* = 0.001), anosmia (*p* < 0.001), asthenia (*p* < 0.001), and diarrhoea (*p* = 0.004) than nursing home patients, while the nursing home patients presented more frequency of dyspnoea (*p* = 0.028) than the community patients. However, there were no significant differences in dyspnoea (*p* = 0.898) between the nursing home patients older than 69 years of age and the community patients older than 69 years of age (Table 2).

The most frequent comorbidities observed in the global sample were arterial hypertension, heart disease, neurological pathologies, and diabetes. In comparison to the community patients, the nursing home patients presented a higher prevalence of arterial hypertension (*p* < 0.001), diabetes (*p* < 0.001), chronic obstructive pulmonary disease (*p* < 0.001), heart disease (*p* < 0.001), chronic renal failure (*p* < 0.001), neoplasm history (*p* = 0.009), and neurological pathologies (*p* < 0.001). However, in comparison to the community patients older than 69 years of age, similarity between both groups was observed. The nursing home patients only presented a significantly higher prevalence of neurological pathologies (*p* = 0.002).

We did not observe significant differences regarding obesity between any of the groups compared (Table 2).

### 3.3. Mortality Analysis

We did not observe significant differences in COVID-19-related mortality between men and women neither in the total population X^2^ = 0.088 (*p* = 2.910), nor in any of the groups studied, including nursing home patients X^2^ = 2.720 (*p* = 0.099), community patients X^2^ = 1.351 (*p* = 0.245), and community patients older than 69 years of age X^2^ = 0.163 (*p* = 0.685).

In the mortality analyses, being over 69 years of age was associated with a higher mortality in all the samples and in the community patients (X^2^ = 28.597 *p* < 0.01 and X^2^ = 18,713 *p* < 0.01, respectively) but not in the nursing home patients (X^2^ = 1.649 *p* = 0.199).

Mortality mainly occurred in nursing homes (80.39%); 19.60% of mortality occurred among community patients, and 15.68% of total mortality occurred among community patients older than 69 years of age (Table 1).

The specific mortality rate (SMR) from COVID-19 was higher in the nursing homes than in the other groups studied (Table 3). The reason for this was the higher COVID-19 incidence in nursing homes. In the incidence analysis, we found a significant association between being infected (yes/no) and living in a nursing home compared to living in the community (X^2^ = 2925.5, *p* < 0.001), or living in the community and being over 69 years of age (X^2^ = 850.635, *p* < 0.001). We did not find this association when comparing living in a community to living in a community and being over 69 years of age (X^2^ = 2.599, *p* = 0.106).

In the mortality comparative analysis between different groups, there were significant differences in mortality when comparing the nursing home patients or the community patients over 69 years of age with the community patients (*p* < 0.001), but not when comparing the nursing home patients with the community patients over 69 years of age (*p* = 0.614). (Table 4).

The case fatality rate (CFR) of the nursing home patients was similar to that of the community patients older than 69 years of age, and very different from the community patients, as we expected when observing the contingency tables of statistical analysis (Table 3).

We also studied the dependency relationship between the different symptoms and comorbidities with mortality. The symptoms associated with higher mortality among all patients included fever, dyspnoea, and asthenia. The comorbidities associated with higher mortality among all the samples included arterial hypertension, diabetes, lung disease, chronic obstructive disease, heart disease, chronic renal failure, history of neoplasms, and neurological pathologies. Obesity was not associated with higher mortality (Table 5).

However, in the comparative study by groups, we found that dyspnoea was the only symptom associated with mortality specifically in nursing home patients, and neoplasm history was the only comorbidity associated with mortality specifically in community patients older than 69 years of age (Table 6).

We also found that hospital admission was associated with lower mortality in the total sample, but not in patients over 69 years of age, neither in those from nursing homes nor in those from the community (Table 7).

## 4. Discussion

Our study, unlike others [19,36,37], shows no differences in mortality by sex during the first wave of COVID-19.

We observed that being admitted in a nursing home is a risk factor for infection. This was expected, since nursing homes are closed areas and residents are exposed to visitors and healthcare workers. Similar results were published in other closed facilities, such as prisons [26,27].

Our results show that most of the patients who died during the first wave were older than 69 years of age and were living in nursing homes. However, in the infected patients, when comparing those older than 69 years of age to COVID-19-related mortality, there was no higher mortality associated with nursing home inhabitants. A study among nursing home populations found higher mortality in the infected COVID-19 patients compared with the residents who were not infected with COVID-19, but not compared with the community population [29]. Another study showed higher mortality in the infected patients over 80 years of age in nursing homes than in the community patients, but only hospitalized patients were studied [36]. Another study showed higher mortality in nursing homes than in a community group control. In this study, the analysis of mortality was carried out by calculating the excess mortality; the fatality rate was not reported, and no statistical analysis was reported comparing whether the higher mortality in nursing homes was statistically significant compared to the control group [19].

According to our results, in COVID-19-infected patients, the mortality directly attributable to COVID-19 was 24.17%, and 80.4% of the total of deaths occurred in nursing homes. In addition, the specific mortality rate was 2.27^0^/_00_ in the total sample and 101^0^/_00_ in nursing homes. These results are different to other published studies, in which the calculation of excess mortality was used [16]; however, our results are consistent with the estimated excess mortality rate for Catalonia obtained by Wang et al. [17]. This last study also obtained the estimated mortality rate results like ours in other areas of Spain such as the communities of Madrid, Castilla-La Mancha, or Castilla-Leon, as well as in other countries such as Italy, Estonia, Serbia, or Ukraine. Unlike these, our study was not carried out from estimates, but from an assessment of each clinical history; we present a real count of all patients with a confirmed or suspected COVID-19 diagnosis that died during the study period.

Like other studies [20,24,25], we found that the following symptoms and comorbidities were associated with a higher mortality in COVID-19 patients: fever, dyspnoea, asthenia, arterial hypertension, diabetes, chronic obstructive pulmonary disease, heart disease, chronic renal failure, history of neoplasms, and neurological pathologies. However, in comparative study by groups, we found that dyspnoea was the only symptom associated with mortality specifically in nursing home patients, and neoplasm history was the only comorbidity associated with mortality specifically in community patients older than 69 years of age. Several studies showed higher mortality in patients with comorbidities, but they did not compare the results with the community population [20,29,30].

In our study, obesity was not associated with higher mortality neither in the total population nor in the different groups in agreement with other published studies [38,39,40].

Hospital admission was associated with a decrease in mortality in the total sample. However, it was not a protective factor for patients over 69 years of age, either living in nursing homes, or in the community, or both.

According to our SMR and CFR results, we can assert that the first wave of COVID-19 had a greater impact on the Balaguer PCC Health Area, in comparison to other territories of Spain [17].

To our knowledge, this is the first study conducted in the primary care setting with a comparative analysis, through a prospective database of mortality, comorbidities, and hospital referral, between nursing home patients and community patients older than 69 years of age, in the same territory, during the first wave. We consider that the main strength of our study is our database. After an extensive bibliographic search, we did not find any study in this scenario around the world whose database has been carried out prospectively.

The main limitation of our study is the size of the sample; however, it reflects what happened during the first wave of COVID-19 in the area of influence of PCC Balaguer with about 28,000 inhabitants. The lack of RT-PCR tests during the first wave makes analysing COVID-19-related mortality as a dependent variable a critical point. This limitation prevented the diagnosis of the asymptomatic population or the population with mild symptoms, which may have led to an overestimation of the case fatality rate. However, we understand that this limitation would not modify the main results of our research, since our study compares different infected population groups (we obtained the RT-PCR results in 96.68% of the sample). Finally, the database originally had an epidemiological–clinical care objective and not a research purpose, and this could have led to some biases. For example, it does not consider the secondary impact (worsening of pre-existing conditions or appearance of new ones).

## 5. Conclusions

The higher mortality rate related to COVID-19 in nursing homes during the first wave of infection was due to a higher incidence compared to the population older than 69 years of age living in the community; however, the fatality rate was similar in both groups. The results of this work are helpful to understand the importance of taking individual measures, such as the correct method of individual isolation. These results may also be helpful in defining future public health decision-making related to contagious diseases; this includes, for example, the administration of booster vaccine doses or new treatments (e.g., prioritizing patients over 69 years of age), and the need for contagion control policies among people older than 69 years of age (e.g., prioritizing the contribution of economic, material, and human resources for the care of the geriatric population). In addition, this study may inform the search for better detection mechanisms for asymptomatic patients, and for the strategies to improve the perception of the risk of contagious diseases that target the population of people over 69 years of age.

## Figures and Tables

**Table 1 geriatrics-08-00048-t001:** Sample distribution.

	Mean Age	Male	Deaths
Total patients COVID-19+ n = 211 (100%)	70	86 (40.75%)	51 (100%)
Nursing home patients COVID-19+ n = 108 (51.18%)	86	42 (38.89%)	41 (80.39%)
Total community patients COVID-19+ n = 103 (48.81%)	53.78	44 (42.71%)	10 (19.60%)
Community patients over 69 years of age COVID-19+ n = 25 (24.27%)	79.04	18 (72.00%)	8 (15.68%)

**Table 2 geriatrics-08-00048-t002:** Distribution and comparative analysis of symptoms and comorbidities between total patients, nursing home patients, community patients, and community patients older than 69 years of age.

Symptom or Comorbidity	Total Patients (n = 211)	Nursing Home Patients (n = 108)	Community Patients (n = 103)	Community Patients over 69 Years of Age (n = 25)	Community Patients in Comparison to Nursing Home Patients. X2 Value and Resulting *p*-Value of This Analysis	Community Patients over 69 Years of Age in Comparison to Nursing Home Patients X2 Value and Resulting *p*-Value of This Analysis
Fever	128 (60.66%)	61 (56.48%)	67 (65.04%)	19 (76%)	1.621 (*p* = 0.202)	3.22 (*p* = 0.072)
Odynophagia	9 (4.26%)	4 (3.70%)	5 (4.85%)	0	0.170 (*p* = 0.679)	0.954 (*p* = 0.328)
Expectoration	5 (2.36%)	4 (3.70%)	1 (0.97%)	1 (4%)	1.701 (*p* = 0.192)	0.004 (*p* = 0.944)
Cough	73 (34.6%)	26 (24.07%)	47 (45.63%)	11 (44%)	10.827 (*p* = 0.001)	4.014 (*p* = 0.045)
Headache	16 (7.58%)	2 (1.85%)	14 (13.59%)	4 (16%)	10.369 (*p* = 0.001)	9.433 (*p* = 0.002)
Dyspnoea	75 (35.54%)	46 (42.59%)	29 (28.15%)	11 (44%)	4.796 (*p* = 0.028)	0.016 (*p* = 0.898)
Anosmia	15 (7.1%)	0	15 (14.56%)	0	16.931 (*p* < 0.001)	0
Asthenia	34 (16.1%)	3 (2.77%)	31 (30.09%)	9 (36%)	29.109 (*p* < 0.001)	27.29 (*p* < 0.01)
Diarrhoea	11 (5.21%)	1 (0.92%)	10 (9.70%)	4 (16%)	8.229 (*p* = 0.004)	12.74 (*p* < 0.001
Obesity	47 (22.27%)	25 (23.14%)	22 (21.35%)	10 (40%)	0.097 (*p* = 0.754)	2.073 (*p* = 0.084)
Arterial hypertension	108 (51.18%)	79 (73.14%)	29 (28.15%)	17 (68%)	42.71 (*p* < 0.001)	0.267 (*p* = 0.604)
Diabetes	58 (27.48%)	44 (40.74%)	14 (13.59%)	9 (36%)	19.497 (*p* < 0.001)	0.190 (*p* = 0.662)
COPD	45 (21.32%)	33 (30.55%)	12 (11.65%)	8 (32%)	11.229 (*p* < 0.001)	0.019 (*p* = 0.887)
Heart disease	62 (29.38%)	46 (42.59%)	16 (15.53%)	11 (44%)	18.602 (*p* < 0.001)	0.016 (*p* = 0.898)
Chronic renal failure	48 (22.74%)	42 (38.88%)	6 (5.82%)	6 (24%)	32.795 (*p* < 0.001)	1.951 (*p* = 0.162)
Neoplasia’s history	22 (10.42%)	17 (15.74%)	5 (4.86%)	3 (12%)	6.689 (*p* = 0.009)	0.309 (*p* = 0.577)
Neurological pathologies	53 (25.11%)	48 (44.44%)	5 (4.86%)	3 (12%)	43.932 (*p* < 0.001)	9.038 (*p* = 0.002)

**Table 3 geriatrics-08-00048-t003:** COVID-19-specific mortality rate and case fatality rate in Balaguer PCC.

Total Population	Specific Mortality Rate	Case Fatality Rate
Total population over 20 years (n = 22.458) Total infected patients over 20 years (n = 211)	2.27^0^/_00_	24.17%
Total population admitted in nursing homes (n = 406) Total infected patients admitted in nursing homes (n = 108)	101^0^/_00_	37.96%
Total population living in community (n = 22.052) Total infected patients living in community (n = 103)	0.45^0^/_00_	9.70%
Total population living in community over 69 years (n = 4005) Total infected patients living in community over 69 years (n = 25)	2^0^/_00_	32%

**Table 4 geriatrics-08-00048-t004:** Mortality comparative analysis between different COVID-19-infected population groups.

Mortality in Interest Group	Mortality in Comparison Group	Chi-Square
Nursing home patients (n = 108) 41	Community patients (n = 103) 10	22.96 (*p* < 0.001)
Nursing home patients over 69 years (n = 107) 40	Community patients over 69 years (n = 25) 8	0.253 (*p* = 0.614)
Community patients under 69 years (n = 78) 2	Community patients over 69 years (n = 25) 8	18.713 (*p* < 0.001)

**Table 5 geriatrics-08-00048-t005:** Association of symptoms and comorbidities with mortality among all patients.

Symptoms and Comorbidities	Total Patients n = 211	Deceased Patients n = 51	Non-Deceased Patients n = 160	Chi-Square n = 25
Fever	128 (60.66%)	37 (72.54%)	91 (56.87%)	3.981 (*p* = 0.046)
Odynophagia	9 (4.26%)	2 (3.92%)	7 (4.37%)	0.019 (*p* = 0.889)
Expectoration	5 (2.36%)	3 (5.88%)	2 (1.25%)	3.587 (*p* = 0.058)
Cough	73 (34.59%)	12 (23.52%)	61 (38.12%)	3.640 (*p* = 0.056)
Headache	16 (07.58%)	3 (5.88%)	13 (8.12%)	0.277 (*p* = 0.598)
Dyspnoea	75 (35.54%)	36 (70.58%)	39 (24.37%)	36.05 (*p* < 0.001)
Anosmia	15 (7.10%)	1 (1.96%)	14 (8.75%)	2.699 (*p* = 0.100)
Asthenia	34 (16.11%)	3 (5.88%)	31 (19.37%)	5.208 (*p* = 0.022)
Diarrhoea	11 (5.21%)	3 (5.88%)	8 (5.00%)	0.060 (*p* = 0.805)
Obesity	47(22.27%)	13 (25.49%)	34 (21.25%)	0.401 (*p* = 0.526)
Arterial hypertension	108 (51.18%)	40 (78.43%)	68 (42.50%)	19.982 (*p* < 0.001)
Diabetes	58 (27.48%)	27 (52.94%)	31(19.37%)	21.86 (*p* < 0.001)
Chronic obstructive pulmonary disease	45(21.32%)	16 (31.37%)	29 (18.12%)	4.045 (*p* = 0.044)
Heart disease	62(29.38%)	23 (45.09%)	39 (24.37%)	8.003 (*p* = 0.004)
Chronic renal failure	48(22.74%)	22(43.13%)	26 (16.25%)	15.908 (*p* < 0.001)
Neoplasm history	22 (10.42%)	13 (25.49%)	9 (5.62%)	16.340 (*p* < 0.001)
Neurological pathologies	53 (21.11%)	21 (41.17%)	32 (2.00%)	9.220 (*p* = 0.002)

**Table 6 geriatrics-08-00048-t006:** Comparative study of symptoms and comorbidities with mortality in nursing home patients and community patients over 69 years of age.

Symptoms and Comorbidities	Association (X^2^) of Symptoms and Comorbidities with Mortality in Nursing Home Patients (n = 108)	Association (X^2^) of Symptoms and Comorbidities with Mortality in Community Patients over 69 Years of Age (n = 25)
Fever	3.750 (*p* = 0.052)	0.853 (*p* = 0.355)
Odynophagia	0.296 (*p* = 0.586)	Insufficient sample
Expectoration	2.419 (*p* = 0.119)	0.490 (*p* = 0.483)
Cough	0.162 (*p* = 0.686)	0.201 (*p* = 0.653)
Headache	0.125 (*p* = 0.723)	0.709 (*p* = 0.399)
Dyspnoea	24.465 (*p* < 0.01)	0.171 (*p* = 0.678)
Anosmia	Insufficient sample	Insufficient sample
Asthenia	1.805 (*p* = 0.179)	0.011 (*p* = 0.914)
Diarrhoea	1.164 (*p* = 0.199)	0.709 (*p* = 0.399)
Obesity	0.053 (*p* = 0.817)	0.490 (*p* = 0.483)
Arterial hypertension	1.812 (*p* = 0.178)	0.163 (*p* = 0.685)
Diabetes	3.005 (*p* = 0.082)	3.585 (*p* = 0.058)
Chronic obstructive pulmonary disease	0.041 (*p* = 0.838)	0.163 (*p* = 0.685)
Heart disease	0.046 (*p* = 0.829)	0.171 (0.678)
Chronic renal failure	1.969 (*p* = 0.160)	1.175 (*p* = 0.278)
Neoplasm history	3.727 (*p* = 0.053)	7.244 (*p* = 0.007)
Neurological pathologies	0.096 (*p* = 0.756)	1.882 (*p* = 0.170)

**Table 7 geriatrics-08-00048-t007:** Association between variables, hospital admission, and mortality in different population groups.

	Hospital Admissions n (%)	Deceased Patients n (%)	X^2^
Total sample (n = 211)	51 (24.17%)	18 (35.29%)	4.540 (*p* = 0.033)
Total patients over 69 years of age (n = 132)	35 (26.51%)	16 (45.47%)	1.799 (*p* = 0.179)
Nursing home patients (n = 108)	19 (17.59%)	10 (52.63%)	2.106 (*p* = 0.146)
Community patients over 69 (n = 25)	16 (64.00%)	6 (37.5%)	0.617 (*p* = 0.431)

## Data Availability

Anonymized data presented in this study are available upon reasonable request. The data are not publicly available due to the sensitivity of the data. Data management has been previously supervised by the Ethics Committee of the Institut d’Investigació en Atenció Primària Jordi Gol i Gurina (Barcelona, Spain) (registration number p21/193-PCV).

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
