# Peer review of "Population Older than 69 Had Similar Fatality Rates Independently If They Were Admitted in Nursing Homes or Lived in the Community: A Retrospective Observational Study during COVID-19 First Wave"

_geriatrics, 2023, doi:10.3390/geriatrics8030048_

Round 1

Reviewer 1 Report

Thank you for the opportunity to review this manuscript.

 In this study, the authors aimed to assess the influence of living in nursing homes on COVID-19.

COVID-19-related mortality and to calculate the real specific mortality rate caused for COVID-19 among people older than 20 years in the Balaguer Primary Care Center Health Area during the first wave of the pandemic. An observational study based on a database generated between 15

March and May 2020

Finally, they concluded Living in a nursing home was associated with a higher incidence of COVID-19 infection but not higher mortality in patients over 69 years (p=0.614). The actual specific mortality rate caused for COVID-19 was 2.270/00. The comorbidities were not associated with higher mortality in the infected nursing home patients group, nor in the infected community patients over 69 years group (except neoplasm history in this last group). 

The introduction needs a research question 

The cited references are relevant to the research. 

The research design is appropriate but needs a structured description of the methods. The description of the data collection should be done in another section after the research design.

First, the design and the participants must be described, and a section must be made that describes the data collection and the variables.

The results are presented clearly, but there is a part explaining that may be poor interrelatedness between items. It does not depend on sample size but on partial correlations, I think I understand, but I was hoping you could explain it better.

The discussion needs to be compared with other territories of Spain and countries. I think the findings are similar to other studies. 

The results support the conclusions.

After controlling the attached paper with minor revisions, 

I recommend that the paper can be published in the Geriatrics journal.  

Author Response

  1. The introduction needs a research question:

Answer: We have rewritten our Ms and included a paragraph expressing that we felt special concern about if living in a nursing home, could be an enough explanatory variable of the high mortality that occurred in those facilities during the first wave of the COVID-19 infection (line 70). 

  1. The research design is appropriate but needs a structured description of the methods. The description of the data collection should be done in another section after the research design.

Answer: We have implemented these changes in the corrected manuscript and a new section of Materials and Methods can be read (line 127).

  1. First, the design and the participants must be described, and a section must be made that describes the data collection and the variables.

Answer: In the corrected manuscript at the beginning of the Materials and Methods explanation a short paragraph about the description of the participants and as has been said in previous point new section of Materials and Methods has been written (line 87 and line 94).

  1. The results are presented clearly, but there is a part explaining that may be poor interrelatedness between items. It does not depend on sample size but on partial correlations, I think I understand, but I was hoping you could explain it better.

Answer: We are sorry, but we don’t understand your question.

  1. The discussion needs to be compared with other territories of Spain and countries. I think the findings are similar to other studies. 

Answer: A wider explanation of reference 17 about mortality rates in other areas of Spain and other countries has been included in the discussion (line 289).

Reviewer 2 Report

1. T firts part of methodology (before study design and participants) should be shorter, especially first paragraph should be shorter.

2. Sample distribution should be mentioned in a study design and participants.

3. I think that tables in the results should be organized with minor changes. For example: colum for total patients n (%), so in the table for Fever of total patients it should be written 128 (60.66). 

4. In table 5 and 6 the percentages should be included for every symptom and comobidity. 

5. In table 7, the percentages should be included, as I explained in a comment number 3.

6. Introduction and discussion are well written.

7. In discussion please explain in more detail limitations of the study, as well as strengths of the study.

Author Response

  1. T firts part of methodology (before study design and participants) should be shorter, especially first paragraph should be shorter.

Thank you for your appreciation. We have rewritten the part the reviewer refers to and explained it in a single section of “Study design and participants” (line 84). 

  1. Sample distribution should be mentioned in a study design and participants.

In line 94 of the corrected manuscript is explines that The final studied sample was of 211 patients with a mean age of 70, of which 86 (40.75%) were male.

  1. I think that tables in the results should be organized with minor changes. For example: colum for total patients n (%), so in the table for Fever of total patients it should be written 128 (60.66). 

We agree that this way of presenting the tables is more attractive and we have made changes in Table 5.

  1. In table 5 and 6 the percentages should be included for every symptom and comorbidity. 

As we just mentioned we have portrayed the percentages in Table 5, we consider not necessary to show the percentage information in Table 6, we consider that it may be redundant.

  1. In table 7, the percentages should be included, as I explained in a comment number 3.

Agreed and done.

  1. Introduction and discussion are well written.

      Thank you very much.

  1. In discussion please explain in more detail limitations of the study, as well as strengths of the study.

Thank you for your assessment. We have some effort on being more explicit about the reflexions of strengths and limitations of our study.

Reviewer 3 Report

Dear Authors

I appreciated reviewing your work.

Your work was well conducted; however, I think the presentation of the results can be improved.

So:

1) In the abstract line 14, check if you want to write "... people older than 20 years" or if it is a mistake.

2) In the "statistical analysis" section, you choose the arithmetic mean. You must prove that the distributions are "normal" to use that measure. So I suggest analysing the distributions' normality and selecting the best measures (mean + standard deviation or median + interquartile range)

3) In Table 2, in the comparison columns, you have a number before the ( ). What is it? Is it the value of the Chi-square test? If yes, please identify it in the first row or at the end of the table.

4) Line 176: you wrote "de sample". I think that you wanted to write "the sample". Please verify.

5) To improve your analysis, I strongly suggest the use of logistic regression

6) Discussion-lines 252-255: you integrate the BMI in the analysis and discussion, but reading your text, it is not clear how you determine that measure

Author Response

1) In the abstract line 14, check if you want to write "... people older than 20 years" or if it is a mistake.

Answer: Thank you for your appreciation. The content of the mentioned sentence is not a mistake, the selected cut-off point at 20 years allows us to study the entire adult population living in the community and to be able to compare the results of our secondary aim (calculating the true specific mortality rate) with similar published studies.

2) In the "statistical analysis" section, you choose the arithmetic mean. You must prove that the distributions are "normal" to use that measure. So I suggest analysing the distributions' normality and selecting the best measures (mean + standard deviation or median + interquartile range)

Answer: The referee is right and we have corrected this sentence in the text to generalize the possible us of centralization measures. In any case, the use of the aritmethic mean has been restricted to identify the average age of the sample distribution.

3) In Table 2, in the comparison columns, you have a number before the ( ). What is it? Is it the value of the Chi-square test? If yes, please identify it in the first row or at the end of the table.

Answer: The reviewer is right. Yes, we have this information in the first row of this table.

4) Line 176: you wrote "de sample". I think that you wanted to write "the sample". Please verify.

Answer: Sorry, verified and corrected.

5) To improve your analysis, I strongly suggest the use of logistic regression

Answer: This is a descriptive work where we want to identify possible tends in the dataset considered. So the use of descriptive tools and the Chi-square test to compare variables are enough for the purpose of this work.  Obviously, the logistic regression modelling framework could also be of interest, though it is not strictly necessary for the purpose of this work.

6) Discussion-lines 252-255: you integrate the BMI in the analysis and discussion, but reading your text, it is not clear how you determine that measure

Answer: The content of lines 305-308 of the Ms is mixed-up. We have decided to rewrite it and to erase “the body mass index” expression.

Reviewer 4 Report

The authors undertook a study of COVID-19-related mortality in elderly people admitted to a nursing home during the 2nd Spanish first wave: a retrospective observational study.

In this context, they aimed to analyse how influenced living in a nursing home (LNH) on COVID-19-related mortality and to calculate the real specific mortality rate caused for COVID-19 (including COVID-19-related mortality not confirmed by RT-PCR) among people older than 20 years in the health area of Balaguer during the first wave, considering several risk factors, in particular if they were admitted in a nursing home.

Please clarify the criterion of age over 20, since the work is assumed to be about older people living in two communities.

Data were collected from two different populations: infected patients living in nursing homes and infected patients living in the community (outside nursing homes). The other study variables were: age; sex; symptoms (temperature above 37ºC, odynophagia, expectoration, cough, headache, dyspnoea, anosmia, asthenia, and diarrhoea); evolution during the month after the confirmation of the infection or the onset of symptoms (cured/deceased); and hospital admission. Comorbidities were collected from the clinical history and included in the database (obesity; hypertension; diabetes mellitus; chronic obstructive pulmonary disease (COPD); cardiovascular disease; chronic kidney failure; neoplasm history; and neurological pathologies).

The final studied sample was of 211 patients with a mean age of 70, of which 86 (40.75%) were male. Of the 211 patients, 108 (51.18%) were admitted in nursing homes; the mean age was 86 years; and 42 (38.89%) were male. The other 103 (48.82%) were living in the community;  the mean age of 53.78 years; and 44 (42.71%) were male.

Authors observed that being admitted in a nursing home is a risk factor for infection. The results show that most of patients who died during the first wave were older than 69 years and were living in nursing homes. However, in infected patients, when comparing older than 69 years COVID-19-related mortality there was no higher mortality associated to nursing homes inhabitants.

The main limitation of the study is the size of the sample; however, it reflects what happened during first wave of COVID-19 in the area of influence of PCC Balaguer with about 28,000 inhabitants. Another limitation is that the lack of RT-PCR tests during the first wave prevented the diagnosis of asymptomatic population or population with mild symptoms, which may have led to an overestimation of the case fatality rate.

The authors concluded that the higher mortality rate related to COVID-19 in nursing homes during the first wave of infection was due to a higher incidence compared to the population older than 69 years living in the community, however, the fatality rate was similar in both groups. The results of this work are important to define public health decisions making like who needs the administration of booster vaccine doses or new treatments (e.g., prioritizing patients over 69 years); the need for contagion control policies among people older than 69 years; the search for better detection mechanisms for asymptomatic patient; and the strategies to improve COVID-19 risk perception.

The work is written very well. Its biggest limitation is the low group size. Perhaps it would be useful to compare morbidity and mortality data in successive waves of COVID-19? Do the authors have such data available? This is, after all, a rerospective paper based on medical data. If the authors have such data it would enrich the quality of the paper and justify the conclusions suggested by the authors.

Author Response

Please clarify the criterion of age over 20, since the work is assumed to be about older people living in two communities.

Answer: We decided to set the cut-off point at 20 years to be able to study the entire adult population living in the community and to be able to compare the results of our secondary aim (calculating the true specific mortality rate) with similar published studies. (lines 87-90).

The work is written very well. Its biggest limitation is the low group size. Perhaps it would be useful to compare morbidity and mortality data in successive waves of COVID-19? Do the authors have such data available? This is, after all, a retrospective paper based on medical data. If the authors have such data it would enrich the quality of the paper and justify the conclusions suggested by the authors.

Answer: We are extremely grateful for this appreciation because it opens a very interesting possibility for us to delve into the epidemiological study of viral pandemics in nursing homes. Unfortunately, we do not have enough time to implement this suggestion in this article.